# The Development of Regional Vessel Traffic Congestion Forecasts Using Hybrid Data from an Automatic Identification System and a Port Management Information System

Joonbae Son [1], Dong-Ham Kim [2], Sang-Woong Yun [2], Hye-Jin Kim [2] and Sewon Kim [3,*]

1   Daeyang AI Center 528, Unmanned Object Major, Sejong University, 209 Neungdong Ro, Kwang Jin Gu, Seoul 05006, Republic of Korea
2   Korea Research Institute of Ship and Ocean Engineering, Daejeon 34103, Republic of Korea
3   Unmanned Object Major, Sejong University, 209 Neungdong Ro, Kwang Jin Gu, Seoul 05006, Republic of Korea
*   Correspondence: sewonkim@sejong.ac.kr; Tel.:+82-10-3123-7176

**Abstract:** The present study proposes a new method that forecasts congestion in the area near a port by combining the automatic identification systems of ships and port management information data. The proposed method achieves 85% accuracy for one-day-long ship congestion forecasts. This accuracy level is high enough to act as a reference value for both manned and unmanned operation situations for autonomous vessels in port areas. The proposed forecast algorithm achieves 95% accuracy when used for a one-hour ship congestion forecast. However, the accuracy of the algorithm is degraded to almost half when the automatic identification system or the port management system is used independently.

**Keywords:** ship AIS (automatic identification system); autonomous vessel; port arrival and departure; vessel traffic congestion; vessel congestion forecast; port MIS (port management information system)

## 1. Introduction

Unprecedented high oil prices and decarbonization issues are arousing interest in solutions for navigation advisories. Voyage assistance applications are not only paramount for intelligent ships but are also necessary for autonomous vessels. There are many previous researches for voyage assistance systems to reduce fuel oil consumption and gas emissions as well as to improve the safety of vessels. Lee [1] and Kim [2] developed an optimal route planning solution that is able to minimize fuel oil consumption and gas emissions in autonomous vessels.

In addition to improving voyage efficiency, attention towards ship safety guidance systems has surged. Huang [3] reviews the state of the art of ship collision avoidance research. Additionally, a number of of machine learning and reinforcement learning technologies have been applied to ship collision research. Zhao [4] applied deep-reinforcement learning technology to ship collisions.

This paper develops a new method that provides a voyage safety index for conventional, intelligent, and autonomous vessels. The proposed method delivers a congestion forecast value at a certain time and in a certain area. This information could be utilized during port berthing and unberthing operations. Autonomous vessels, in particular, could use this index as the trigger value that can start port arrival or departure.

According to Anders's [5] work, ship congestion can be defined and forecast using automatic identification data from ships. Ship congestion refers to the intensity of vessel traffic intensity, and the VTS (vessel traffic center) value operator regulates it. For example, the vessel traffic operator conventionally (as is the case at present) manages the vessel traffic that can be maintained at a certain level. If the vessel intensity increases, then it can

cause complex vessel maneuvering situations as well as collision situations. Therefore, this research provides a forecast of the congestion information that can be used for the vessel traffic center operator and autonomous vessels to avoid complex maneuvering situations and bottlenecks, which increase the difficulty of berthing and unberthing.

This manuscript proposes a port congestion estimation method that superposes the estimated route of a vessel obtained from ship AISs (Automatic Identification Systems) and port registration systems (i.e., port management information system). Ship AISs and port registration have a commonality in that they have resource information on the voyage context. Primarily, a ship's AIS records the vessel's movement, so the AIS stores the vessel's speed, position, heading, and engine status every second. Therefore, AIS information is the main resource for ship voyage patterns near the shore.

AISs contain almost all of the information that is essential to analyze vessel voyages. Therefore, many studies have been conducted on the use of AIS data for ship voyage analysis. In a representative work, Zhang [6] reproduced ship trajectories using AIS data. Zhou [7] and Wang [8] proposed a grouping method using AIS data.

AIS data are a central resource that perceives vessel traffic in the port area. Zhang [9] analyzed the spatial and the temporal vessel movement in the Singapore area. Chen [10] and Toscano [11] evaluated the gas emissions of Tijan and Naples by estimating the amount of gas emissions according to vessel movement obtained from the historical AIS data. Zhou [12] obtained the vessel traffic statistics of Rotterdam. However, there is no prior reference that instantaneously considers both AIS and the port-plan reporting system data. Brian [13] used historical AIS data from ships to predict ship behavior and created a deep learning network that learns the past movements of a vessel and predicts the future voyage patterns of a single ship. AIS data were used as the basis for a deep learning network.

Port voyage plan reporting systems have rarely been researched. Kim [14] used the port report system as a database to evaluate gas emissions in the Busan port. Choi [15] proposed a way for electric documents to be interchanged with the port's data management system. A port's registration system is the arrival and departure system and is mandatory when a vessel wants to travel inbound and outbound from a certain port. The data reporting system in a port has the vessel's arrival and departure plan. These data provide information that recognizes the estimated time of arrival (ETA), estimated time of departure (ETD), and berth location. That information delivers the framework of the voyage to the forecast model.

The main contribution of this manuscript is its proposal of a new method that combines the AIS data from a ship and the corresponding port MIS data to create a high-performance ship congestion forecast algorithm. This AIS–port registration hybrid method provides more accurate forecast results compared to AIS or port registration data applied independently. The aim of this manuscript is to propose an estimation algorithm to determine vessel traffic congestion by combining ship AIS data and the vessel's voyage registration information. This research firstly combines the port MIS data and the real-time AIS data for a congestion forecast. Previous research by Zhang et al. [9], Chen et al. [10], and Kim et al. [14] analyzed only the historical AIS movement. Meanwhile, this manuscript proposes a congestion forecast method that adopts the historical AIS data, the port MIS data, and the real time AIS data. The accuracy of the congestion forecast is improved by employing the proposed method.

Vessel traffic bottleneck information is vital not only for autonomous vessels but also for conventional and vessel traffic center operators. Autonomous vessels and remote operation vessels can make voyage decisions based on vessel congestion.

Position and geography, such as the berthing place and the anchorage, are pre-defined before analysis. Figure 1 presents the target research area: South Korea, Ulsan Port.

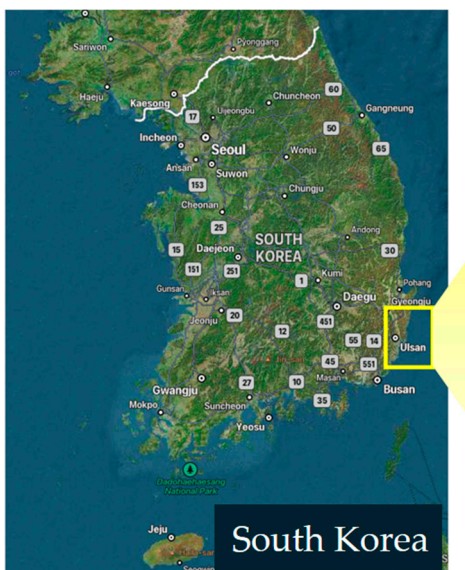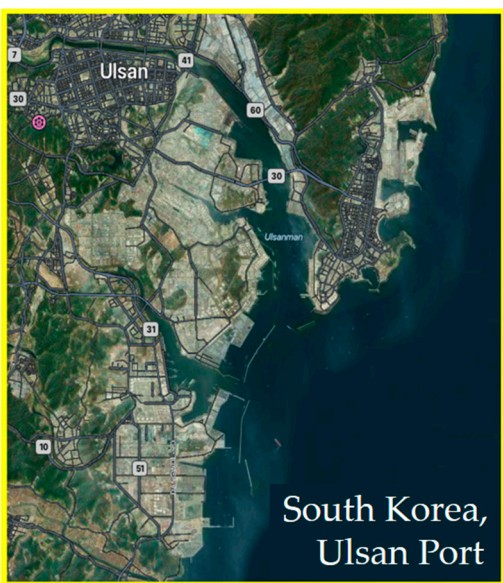

**Figure 1.** Ulsan Port—target site for congestion forecast.

This study used 2020 AIS and port-reporting data from the Ulsan port. Ulsan is a complex port that has 20 berths and various cargo types, such as containers, bulk, cars, and oil. Moreover, it is an area where autonomous vessels perform trials. Ship congestion information is necessary to maneuver in a complex port of this kind in South Korea.

## 2. Materials and Methods

### 2.1. Problem and Materials

2.1.1. Problem Definition

The developed algorithm aims to forecast the congestion in the area at a certain time.

$$C(A_k, t) \in \{low, medium, high\} \tag{1}$$

where $C(A_k\ t)$ = congestion in the area $(A_k)$ at time (t);

$$
\begin{aligned}
C(A_k, t) &= low \ when \ Vn < 3; \\
C(A_k, t) &= medium \ when \ 3 \leq Vn < 5; \\
C(A_k, t) &= high \ when \ 5 \leq Vn < 6.
\end{aligned} \tag{2}
$$

(Standard congestion is based on local Ulsan-area traffic.)

Figure 2 presents the area $(A_k)$. The area is modeled by considering the vessel traffic density and berth places in the Ulsan Port. The Area $(A_k)$ is the district that is supervised by the local vessel traffic center officer. The congestion forecast result is also useful to the vessel traffic center's operator for vessel traffic supervision work. In addition, the Area has the tree structure. The root of the tree is the boarding place of the pilot: the edge point of the $A_1$. The Area's dimension is defined based on the standard that is larger than the vessel traffic kernel density as 0.01. According to Wang's work [8], the vessel traffic density is able to be described by the kernel density function that is the continuous probability function of ship AIS data.

$$Acc(t) = \frac{True \ Positive \ Number \ of \ C(A, t)}{N} \tag{3}$$

Here, Acc(t) is the congestion forecast accuracy at time (t), N is the number of areas (13), and C(A,t) is the congestion in the area and a certain time.

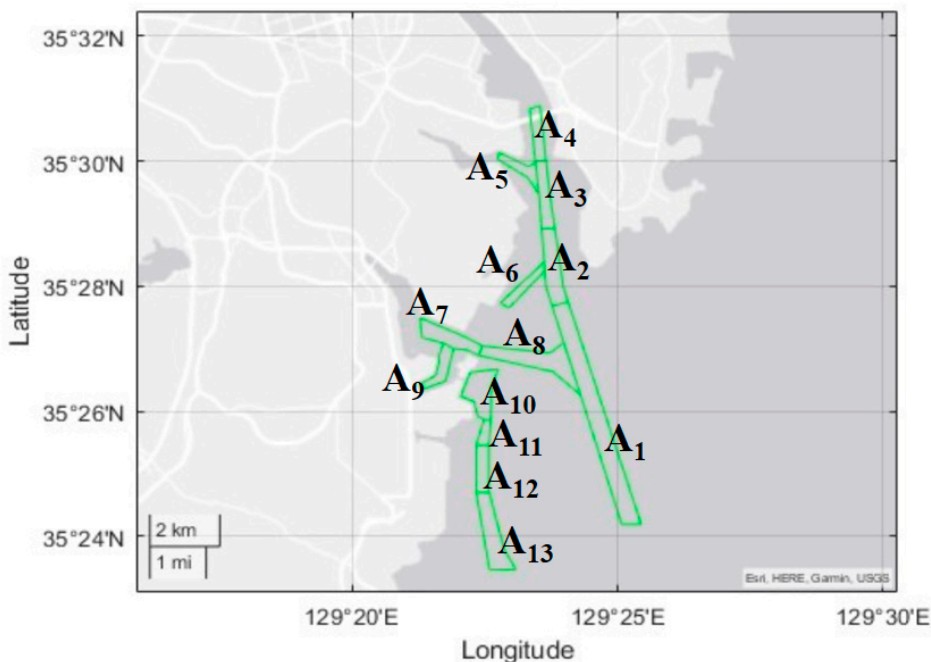

**Figure 2.** Definition of congestion area $A_k$.

2.1.2. Materials

The main contribution of this research is the development of a new ship congestion forecast algorithm obtained by combining AIS data, port registration data, and geographical information. In Table 1, the data categories and data periods are presented. The AIS data were obtained from the ships in the Ulsan Port's arrival and departure information. A port MIS (management information system) is used to declare a vessel's arrival and departure to the government. It is an identical system to the port registration system.

**Table 1.** Input data for the forecast algorithm.

| Data | Period |
|---|---|
| AIS [1] | 2020.01~2020.12 |
| Port MIS [2] | 2020.01~2020.12 |
| Geography Information | Static Information |

[1] Automatic identification system. [2] Port management information system.

The algorithm builds the forecast model using dynamic voyage information from the ship's AIS. Table 2 shows the AIS data feature that was employed to build the forecast model and to assess the performance of the method.

**Table 2.** Ship AIS data.

| Data | Example | Unit (Type) |
|---|---|---|
| Position | 36.25334 N, 129.24391 E | deg |
| SOG | 10 | Knot |
| COG | 30 | deg |
| MMSI | 440055930 | (Integer) |
| CALLSIGN | H9AO | (String) |

Table 3 shows the data feature of the port MIS that was employed to build the forecast model and to assess the method's performance.

**Table 3.** Port MIS data.

| Data | Example | Type |
|---|---|---|
| Callsign | H9AO | String |
| Inbound/outbound | Inbound | String |
| ETA (Estimated Time Arrival) | 1 September 2020 10:00 | Time |
| ETD (Estimated Time Departure) | 4 September 2020 15:01 | Time |
| Berth Place | Mb2 | String |
| Next Port | KRBUS | String |
| Previous Port | KRGSN | String |

*2.2. Methods*

2.2.1. Methodology—Overview

Figure 3 presents the schematic diagram of this methodology. The proposed method consists of two parts: model development and forecast model application. The goals of model development are to regulate data and to make a forecast model. Forecast model application is the process that involves using the forecast model for ship congestion estimation.

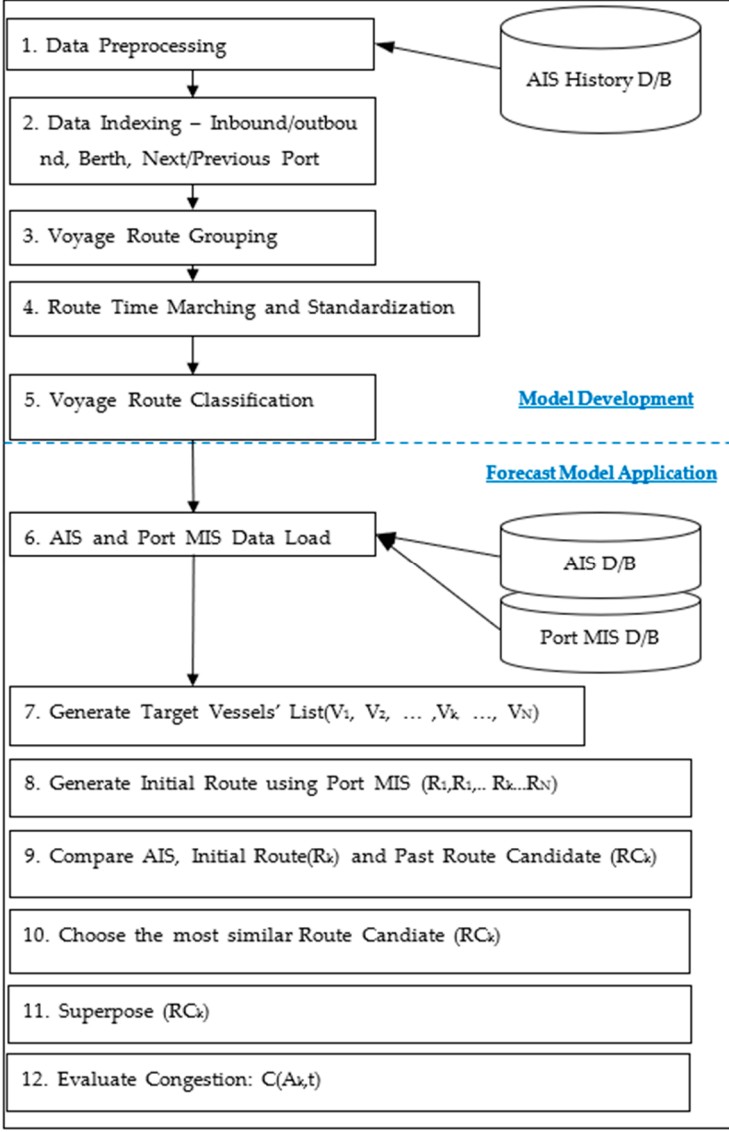

**Figure 3.** Schematic diagram—overall research procedure.

2.2.2. Data Processing

Figure 4 presents the data pre-processing process used for model development. Pre-data processing consists of five steps. The first step is data clean-up, which fills the null data, expels the outliers, and filters by uniform frequency. Then, the area of interest is defined based on the voyage density function and pilotage regulation. In the third step, data are regulated according to the MMSI and Call Sign. After that, the algorithm puts the tags denoting the inbound and outbound voyages. This step aims to split the AIS data into segments that can be analyzed. The last step groups AIS data based on similarities. It categorizes the representative routes for each vessel. Through this step, the forecast can create a route that relates to what actions the ship is carrying out in the present moment.

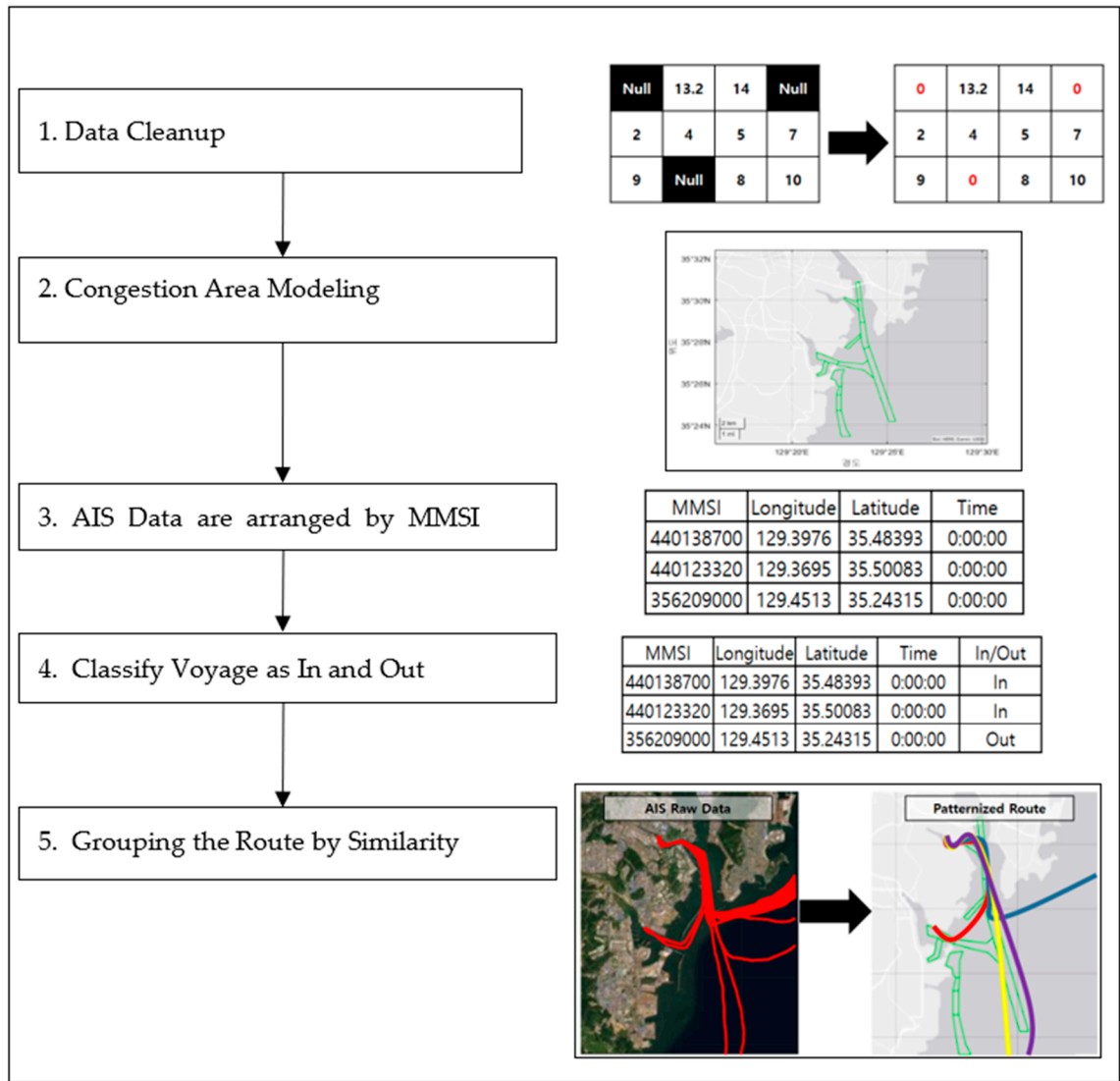

**Figure 4.** Flow of the data processing method.

In order to predict the degree of congestion, AIS data cleaning should be carried out. For that purpose, it is necessary to organize past AIS voyage data according to the AIS route and to store it in a database. Pre-processing of the voyage-route groupings is necessary to classify the track data by voyage and to divide the trajectory into arrival and departure information and to store it.

Ship AISs store information according to the ship's MMSI number. A port registration system stores the data according to the ship, but data are organized according to call signs. Therefore, the key data between ship AIS and the port registration DB are depicted

according to MMSI, as presented in Figure 5. To achieve this, the authors matched the ship AIS information according to the ship MMSI numbers. Thus, it was necessary to link the two databases by using the ship MMSI numbers (call sign) as the database key.

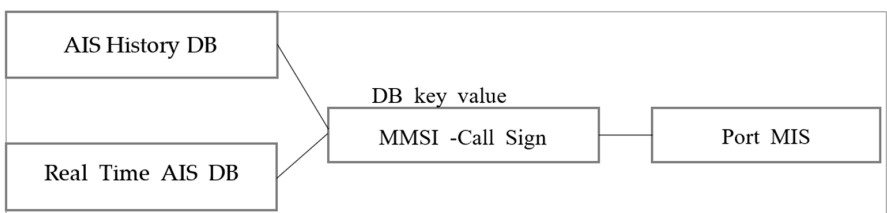

**Figure 5.** Database key between AIS database and Port MIS database.

Figure 6 depicts an overview of the proposed ship congestion algorithm. The forecast algorithm works according to the following process. The algorithm gathers ship AIS and port MIS data every ten minutes. After that, the analyzed target vessel list is generated, and ships are classified as inbound and outbound vessels.

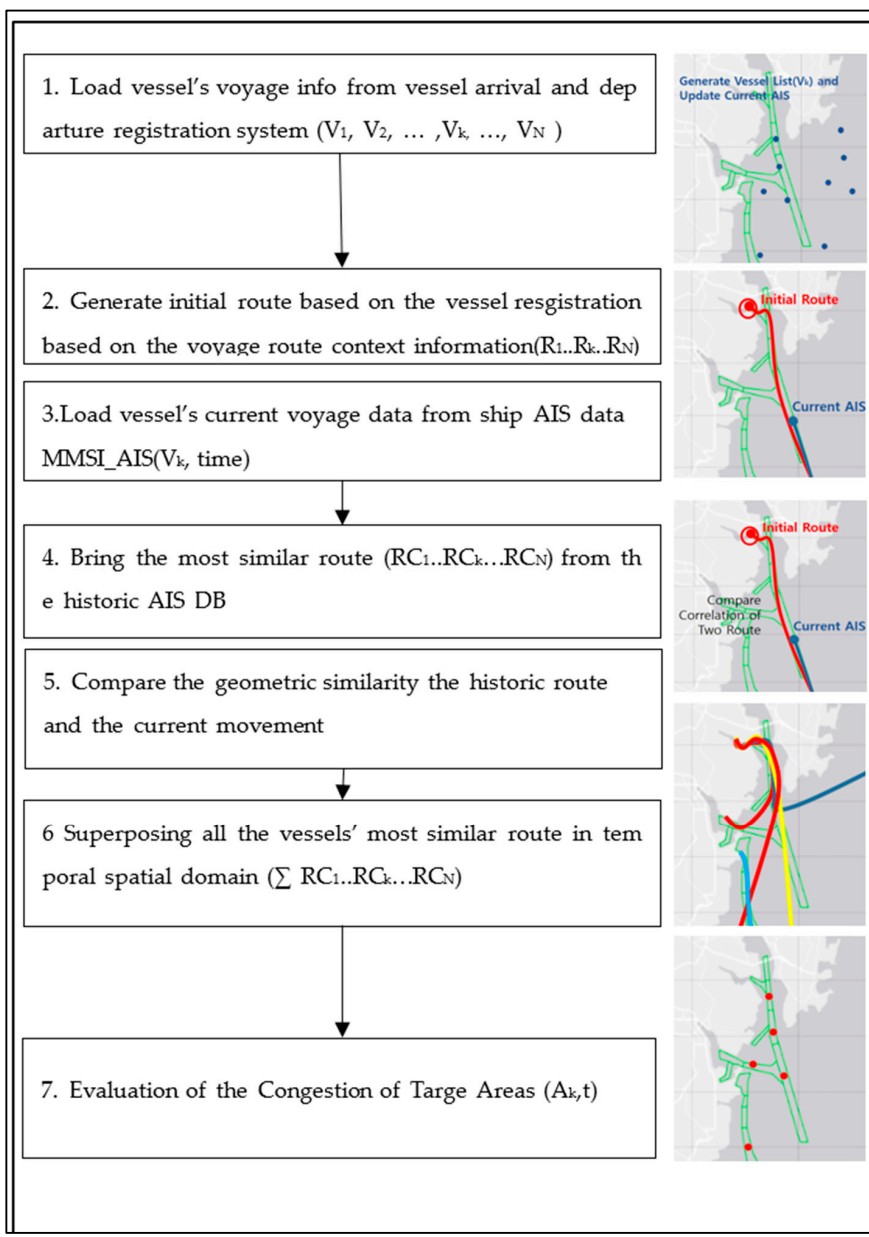

**Figure 6.** Congestion forecast process.

Then, the algorithm loads ship AIS and port MIS information for the current time. In the case of port MIS, the data for one week are loaded because the intervals between ships' arrivals and departures vary. In 98% of cases, the vessel's interval between arrival and departure is shorter than seven days. Therefore, one week is a long enough period. After that, the vessels are classified according to their arrival and departure. This process aims to mark the voyage context in the voyage data.

In the third step, the algorithm brings up the vessels' past routes and uses the most similar route to the present one by considering the voyage context information. Then, the algorithm produces the initial congestion estimation. After the target vessel is newly observed in the AIS data, the algorithm monitors the movement of the vessel. As the vessel voyage progresses, the algorithm compares its similarity with the past routes included in the historic AIS DB. The algorithm updates the route to the one that is the most similar to the current movement of the vessel. Finally, the algorithm accumulates each vessel's voyage route spatially and temporally.

## 3. Results

The ship congestion forecast was evaluated from various perspectives. First, the one-day forecast performance was analyzed. The algorithm calculated the ship congestion within one day—24 h. The authors performed estimates for the 30 days in September and averaged the accuracy. This experiment determined the performance of the one-month ship congestion forecast. The next experiment was designed to investigate the changes in the forecast performance of the algorithm over time. This experiment could determine this algorithm's forecast data for a certain length of time. The third experiment, described in Section 3.3, analyzes the contribution of historical AIS data to the accuracy of the ship congestion forecast. The experiment predicts two different ships: the vessels with AIS voyage data and the vessels that visit the Ulsan Port first and thus have no AIS voyage data. Finally, Section 3.4 explores the involvement of the port reporting system's voyage data in the forecast performance.

### 3.1. One-Day Congestion Forecast Result

The forecast algorithm achieved average 85.0% accuracy and 0.8% standard deviation for a one-day period average by using one-month validation test. Figure 7 shows an example of the comparison between estimation and validation. The color of the outline of the area shows that the congestion within the green line is good. The yellow area has some slow traffic, and it is estimated that there may be a vessel traffic jam in the red area. The performance of the suggested algorithm is excellent. This accuracy indicates that the mode would be a good resource to support the current VTS operator and the autonomous vessels. Table 4 presents congestion accuracy and standard deviation for the area.

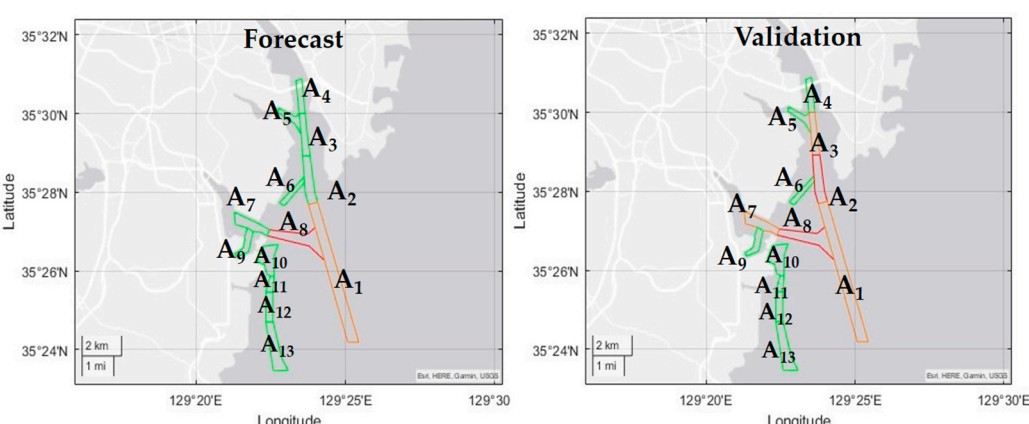

**Figure 7.** Ship congestion forecast results.

**Table 4.** Congestion Forecast Result (1 month period).

| Area ($A_k$) | Accuracy Average (%) | Accuracy Standard Deviation (%) | Area ($A_k$) | Accuracy Average (%) | Accuracy Standard Deviation (%) |
|---|---|---|---|---|---|
| $A_1$ | 85.0 | 0.9 | $A_8$ | 85.0 | 0.8 |
| $A_2$ | 84.0 | 0.9 | $A_9$ | 85.1 | 0.8 |
| $A_3$ | 84.9 | 0.9 | $A_{10}$ | 85.0 | 0.7 |
| $A_4$ | 85.0 | 0.8 | $A_{11}$ | 84.9 | 0.6 |
| $A_5$ | 85.2 | 0.7 | $A_{12}$ | 85.0 | 0.8 |
| $A_6$ | 84.9 | 0.8 | $A_{13}$ | 85.3 | 0.8 |
| $A_7$ | 85.1 | 1.0 | - | - | - |

### 3.2. Forecast Performance by Time Lapse

The main performance index of the suggested forecast algorithm is how much change is observed in the forecast accuracy over time. The forecast algorithm shows a lower limit of accuracy of 92.2% and upper limit of 95.5% accuracy with 95% confidence level when the forecast duration is 1 h, but 24 h later, the accuracy decreased to a lower limit of 78.2% and upper limit of 81.4% with 95% confidence level. The degradation of the forecast algorithm is caused by the availability of AIS data. Ship AIS data within the range of 100 km could be observed, which is under the 4-h range when considering the size of the Ulsan Port. Therefore, the decrease in the forecast accuracy due to the time lapse comes from the AIS range presented in Figure 8.

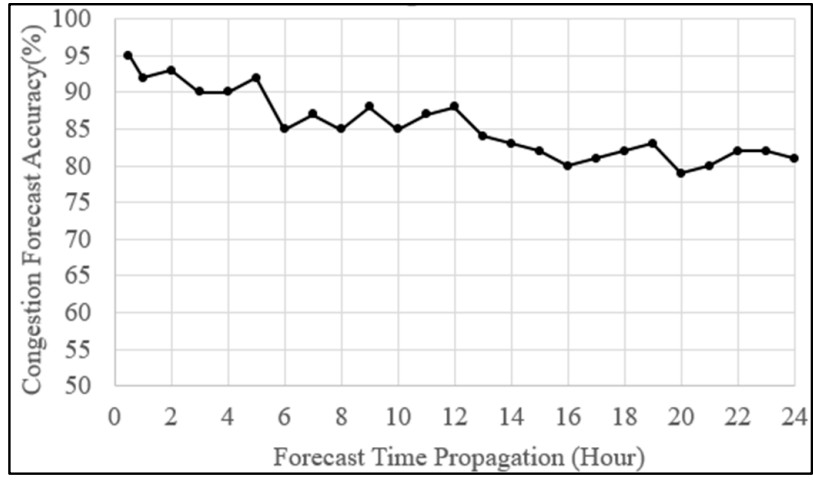

**Figure 8.** Accuracy variations in the forecast due to time propagation.

Meanwhile, an accuracy level of 80% is also high enough to be utilized in autonomous vessel berthing, and for decision making during unberthing operations, the VTS operator can be used.

However, the only context information which the port registration system provides is overall (brief) insights into ship congestion. Therefore, an autonomous vessel could reference the port registration voyage context information when making decisions during berthing and unberthing procedures.

### 3.3. The Effects of a Ship's Voyage History on the Congestion Forecast

A vessel visiting a port for the first time will have no past voyage records. Therefore, this research assumes that these ships would adopt the average route according to ship type. Even though some vessels were completing their first voyage to the port, the suggested algorithm only deduced 7% of the prediction performance, as presented in Figure 9.

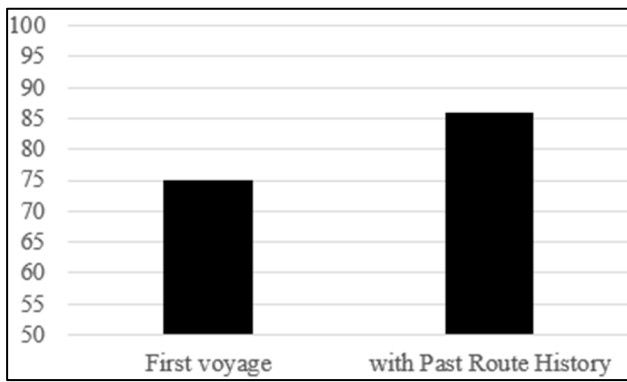

**Figure 9.** Accuracy Variations in the Forecast due to time Propagation.

### 3.4. AIS Historic Contribution

Figure 10 compares the prediction accuracy of the port data combined with AIS data and using only AIS data. The prediction accuracy when the algorithm used the port data and the AIS data together was 40% more accurate than when AIS data were used alone. Therefore, combining ship AIS and port registration data is valuable to make predictions regarding the future locations of vessels.

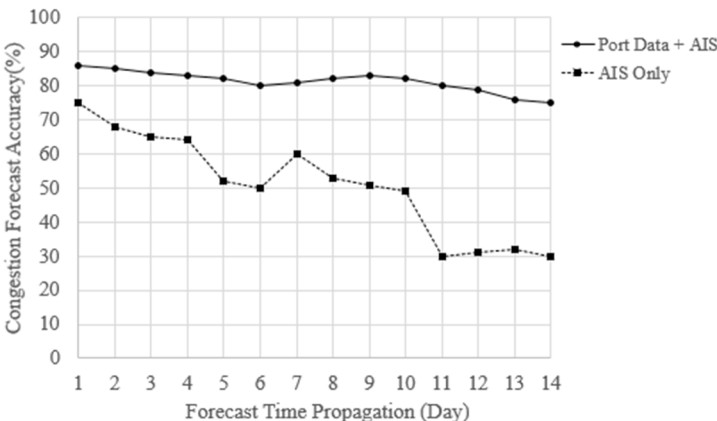

**Figure 10.** Variations in accuracy when using AIS data only and the hybrid case.

### 4. Discussion

The present study proposes a new method to forecast congestion in the area near a port. This study combines ship AIS and port registration information and applies it to voyage context modeling. Additionally, this method proposes a scheme that classifies past AIS data and compares the present AIS data.

The benefit of this manuscript's forecast algorithm to provide congestion forecasts that support the arrival and departure process for both manned and unmanned operation vessels. The VTS operator can utilize the forecast results not only as the conventional vessel traffic regulation method based on the present congestion data but also using the predicted congestion for vessel traffic control. This forecast prediction can also provide a basis for calculating the vessels' fuel oil consumption and the gas emission of the vessels near to shore, as well as to guide port development, and the ship berthing-pilotage planning.

By using this method, an overall accuracy of 85% is achieved for a one-day prediction. This result shows that the model performs well enough to be used when autonomous vessels are making decisions when approaching a port.

After inventing the forecast method, this study explored the method's applicable time-length and the contribution of the AIS and port reporting system data when making predictions regarding ship congestion. The performance of the forecast method was 95%

for a period of one hour, but it decreased by about 80% within 24 h. This reduction phenomenon is caused by the AIS data's real-time value, which is stored within a limited time length.

The contribution of past AIS data resulted in a change in accuracy of about 30%. This value was averaged using one-month data. Port reporting data makes a contribution of about 40% to the ship congestion forecast because it includes voyage planning context information that improves the ship congestion forecast. Moreover, the port reporting data shows the shipping forecast for a longer period of time, such as two weeks, compared to the use of AIS data only.

In future work, it will be worth using this machine learning technology to group historic AIS data to generate past voyage patterns.

**Author Contributions:** Conceptualization, J.S. and S.K.; methodology, J.S. and S.K.; validation, S.-W.Y., D.-H.K. and H.-J.K.; formal analysis, S.K.; data curation J.S. and S.K.; writing—original draft preparation, S.K.; writing—review and editing, S.K.; funding acquisition, S.K. All authors have read and agreed to the published version of the manuscript.

**Funding:** This research was supported by a grant from National R&D Project "Development of Smart Port-Autonomous Ships Linkage Technology" funded by the Ministry of Oceans and Fisheries, Korea (1525012520).

**Conflicts of Interest:** The authors declare no conflict of interest.

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
