# Peer review of "The Development of Regional Vessel Traffic Congestion Forecasts Using Hybrid Data from an Automatic Identification System and a Port Management Information System"

_jmse, doi:10.3390/jmse10121956_

Round 1

Reviewer 1 Report

This paper deals with the method that provides the congestion evaluation of the target site by using AIS and port reporting data. While I appreciate the effort of the work presented, the authors need to improve the paper's focus for publishing as a journal paper.

The authors should clarify the main issue and contribution of this paper compared to other methods. The proposed method is practical, but in other words, I cannot find any novelty of your proposed method.

In line 107-112, this paper classified the congestion by low, medium, and high by using the number of vessels. However, This congestion index is highly dependent on the size of the area. In this paper, the way to determine Ak in Figure 2 can be arbitrarily determined, and the results can vary greatly depending on this determination. Therefore, the proposed method is not acceptable as a generally applicable method.

In chapter 3, the authors discuss the accuracy of the proposed method by using an average index. However, to evaluate whether the proposed method is suitable or not, authors should analyze the individual cases, not averages. I think that there should be some areas that are easy to predict and some areas that are difficult to predict.

Author Response

Dear Sir 

First of all, we sincerely express thanks for your efforts in reviewing our paper. Also, the authors appreciate the reviewer’s insightful comments. The corresponding responses are colored in light blue. We found the reviewer’s comments very helpful in improving the overall quality. As you can see, we have no problem in revising the paper considering all the reviewer’s points. The point-by-point reply is as follows: The revised portion of the paper is marked in red italic. (please refer the attached responses) 

Reviewer 2 Report

The tech note present a method to identify the regional vessel traffic congestion. The method is sound and it contains some result validated the method. I believe it worth to be published in the Journal of Marine Science and Engineering

Author Response

First of all, we sincerely express thanks for your efforts in reviewing our paper. Also, the authors appreciate the reviewer’s insightful comments. We found the reviewers’ comments very helpful in improving the overall quality.

Point 1: The tech note presents a method to identify the regional vessel traffic congestion. The method is sound and it contains some result validated the method. I believe it worth to be published in the Journal of Marine Science and Engineering

Response 1: The manuscript got proofread by a native speaker. The authors revise the manuscript according to the proofread result.

Reviewer 3 Report

The authors propose the combination of AIS and port MIS information in order to predict the congestion in a harbor. Their approach is applied in real life (historical) data and the results are promising.

The core content is fine, the following comments are solely about its presentation.

English is in some parts rather poor, proof reading by a fluent speaker would be beneficial.

In table Table 3 H9AO is not an integer. Perhaps the data type is string?

Results: The average accuracy alone is not a sufficient measure. Confidence intervals and/or standard deviation values could provide a more clear indication of how well the prediction algorithm performs. This also applies in the time lapse, where figure 8 is now misleading (without confidence intervals it appears as if in some cases accuracy increases when time distance increases, which is not rational).

There are two sections numbered as 3.2

The discussion regarding the value/implications of the work can be enhanced. Who will use the information generated from such a predictive system? How? Which will be the benefits in safety, costs and environmental impact? The authors have made a brief mention of the topics in the introduction, but perhaps a deeper view in the discussion section would be useful.

Author Response

Dear Sir 

First of all, we sincerely express thanks for your efforts in reviewing our paper. Also, the authors appreciate the reviewer’s insightful comments. The corresponding responses are colored in light blue. We found the reviewers’ comments very helpful in improving the overall quality. As you can see, we have no problem revising the paper considering all the reviewer’s points. The point-by-point reply is as follows: The revised portion of the paper is marked in red italic.

(Please refer the attachment) 

Round 2

Reviewer 1 Report

The manuscript has been much improved and is in a nice condition now.